# Monocular-Vision-Based Precise Runway Detection Applied to State Estimation for Carrier-Based UAV Landing

**DOI:** 10.3390/s22218385

**Published:** 2022-11-01

**Authors:** Ning Ma, Xiangrui Weng, Yunfeng Cao, Linbin Wu

**Affiliations:** College of Astronautics, Nanjing University of Aeronautics and Astronautics, Nanjing 210016, China

**Keywords:** UAVs, state estimation, monocular vision, runway detection

## Abstract

Improving the level of autonomy during the landing phase helps promote the full-envelope autonomous flight capability of unmanned aerial vehicles (UAVs). Aiming at the identification of potential landing sites, an end-to-end state estimation method for the autonomous landing of carrier-based UAVs based on monocular vision is proposed in this paper, which allows them to discover landing sites in flight by using equipped optical sensors and avoid a crash or damage during normal and emergency landings. This scheme aims to solve two problems: the requirement of accuracy for runway detection and the requirement of precision for UAV state estimation. First, we design a robust runway detection framework on the basis of YOLOv5 (you only look once, ver. 5) with four modules: a data augmentation layer, a feature extraction layer, a feature aggregation layer and a target prediction layer. Then, the corner prediction method based on geometric features is introduced into the prediction model of the detection framework, which enables the landing field prediction to more precisely fit the runway appearance. In simulation experiments, we developed datasets applied to carrier-based UAV landing simulations based on monocular vision. In addition, our method was implemented with help of the PyTorch deep learning tool, which supports the dynamic and efficient construction of a detection network. Results showed that the proposed method achieved a higher precision and better performance on state estimation during carrier-based UAV landings.

## 1. Introduction

A carrier-based unmanned aerial vehicle (UAV) is one type of shipboard aircraft on aircraft carriers, destroyers or other ships, which play irreplaceable roles in maritime security and interests [1]. With the development of modern military technology, carrier-based UAVs can perform a large number of missions in modern warfare, such as reconnaissance and surveillance, fire guidance, target indication and combat damage assessment. Carrier-based UAVs have superior ISR (intelligence, surveillance and reconnaissance) capabilities and the advantage of avoiding casualties. As a flexible, efficient, low-risk and all-weather maritime perception and combat platform, having carrier-based UAVs improves the combat capability of aircraft carriers and enriches the means of maritime composition of combat units.

The landing phase is the most risky and complicated phase of flight and the ability to land autonomously in the face of an emergency should be an irreplaceable part of the deployment of UAVs [2]. According to statistics, more than 30% of carrier-based aircraft flight accidents occur in the landing stage, which is three to six times higher than ground-based landings. Compared with the landing of manned aircraft, the lack of “human-in-the-loop” real-time monitoring and intervention poses a great challenge to the landing of carrier-based UAVs. Additionally, the general requirements for the development of autonomy put forward the functional requirements for autonomous landing and stipulate the accuracy of state estimation during the landing phase. This will undoubtedly help to improve the full-envelope autonomous flight capability of UAVs [3]. At the same time, it poses a severe challenge to the state estimation technology in the landing phase.

According to their airfoil, carrier-based UAVs are divided into fixed-wing UAVs and rotary-wing UAVs. The VTOL (vertical takeoff and landing) UAVs can take off and land vertically on the after-deck using a “Harpoon” or “Bear Trap”. VTOL UAVs have the capability of vertical taking-off and landing, but the shortcomings of a short range and slow speed limit the application in surface reconnaissance and combat [4]. Although fixed-wing UAVs have advantages in terms of payload and endurance, they still face the following challenges to achieve autonomous landing [5]:Since the mass of fixed-wing UAVs is generally much larger than that of VTOL UAVs, traditional recovery approaches such as net recovery and parachute recovery are not applicable, so the landing of fixed-wing UAVs requires the help of a runway;The landing speed of fixed-wing UAVs is so high that there is little room for correction in the event of mistakes, so the fault tolerance of the landing of fixed-wing UAVs is much lower than that of manned aircraft;Fixed-wing UAVs usually land on the deck by means of impact. Such a landing method not only has higher requirements on the strength of the airframe and landing gear, but also puts forward more reliable and accurate requirements for the landing guidance system, which plays a key role in safety.

The landing process of carrier-based UAVs is a relay guidance process. Modern aircraft carriers are equipped with an air traffic control system, TACAN and landing guidance system. The landing guidance system mainly uses radar, satellite, photoelectric and other technical means to provide guidance information for the landing stage of the UAV. The landing guidance information for UAVs are provided by a landing guidance system using radar, GNSS, photoelectric and other technical pieces of equipment [6]. At present, the main UAV landing guidance systems are: the sea-based Joint Precision Approach and Landing System (JPALS), the UAV Common Automatic Recovery System (UCARS), SADA (Système d’Appontage et de Décollage Automatique) and the DeckFinder system [7,8,9]. In order to reduce the landing risk and improve the utilization rate of carrier-based UAVs in the electromagnetic interference environment, it is important to increase the redundancy and release the high dependence on the aircraft-to-carrier data link.

With the rapid development of machine vision technology, vision-based landing guidance has become a research focus in the autonomous landing of carrier-based UAVs [10]. Machine vision comprehensively processes the image information collected from cameras and solves the navigation parameters, such as the position and attitude of the UAV relative to the landing area, without human involvement in decision-making [11]. Compared with traditional guidance methods, machine vision has the following advantages:It has good SWaP (size, weight and power) characteristics [12]. Machine vision sensors are small in size, light in weight and low in power consumption;It has better antistealth and anti-interference abilities [13]. Machine vision typically employs passive autonomous sensors that operate in bands far away from the frequency range of electronic countermeasures. Therefore, it still has measurement capabilities in complex electromagnetic environments, radio silence conditions and satellite-denial environments.Vision information is higher-dimensional. Machine vision can obtain higher-dimensional information, such as battlefield damage assessment, which can assist carrier-based UAVs in dealing with more emergencies.

Although visual navigation has become a promising method for landing systems, tracking the targets continuously and extracting the coordinates of the landing site accurately are challenges for visual algorithms due to the dramatic radial distance changes and complex environment during carrier landing [14]. In addition, the landing area of an aircraft carrier is extremely limited, which is approximately 1/10 of a ground-based runway. The landing requirements are even more demanding for fixed-wing carrier-based UAVs. Referring to the standard of the F/A-18A carrier-based automatic landing system [15], the standard for the vertical average error of fixed-wing UAV landing is ±4.88 m and the standard for the lateral average error is ±1.22 m. Therefore, the accuracy of the pose measurement is required to be higher in the UAV landing stage.

In this paper, an end-to-end state estimation method for autonomous landing of fixed-wing UAVs based on monocular vision is proposed to promote the development of the autonomous landing of carrier-based UAVs. To meet the requirement of accuracy for runway detection and the requirement of precision for UAV state estimation, a runway detection framework is designed including four modules: a data augmentation layer, a feature extraction layer, a feature aggregation layer and a target prediction layer. The main contribution of this paper can be stated as:Different from the traditional two-stage state estimation method including region detection and edge extraction, this paper proposes an end-to-end runway precise detection framework for state estimation, which can effectively improve the calculation efficiency and precision, and is easy to be realized in engineering.This paper introduces the appearance feature of the runway and polygonal prediction information of the runway into the design of the data augmentation layer, which can improve the robustness and accuracy of the runway detection under the rotating perspective.A corner prediction method based on geometric features is introduced into the runway detection framework, which improves the precision and fitness of the landing field prediction. On this basis, the reliability of UAV’s state estimation is also be improved.

The rest of this paper is arranged as follows: in Section 2, related works are discussed; in Section 3, the design and improvement of the network framework are illustrated; in Section 4, the corner prediction method based on geometric features and the design of the loss function are discussed in detail; in Section 5, the datasets for the simulation and experimental setup are described, and the experimental results are evaluated and analyzed; in Section 6, summarizes this paper.

## 2. Related Works

At present, most research teams mainly adopt two ideas to carry out research on vision-based UAV autonomous landing on a carrier. Some researchers design cooperation markers with special geometric shapes, such as “T”, “H”, circle or rectangle, with further auxiliary information such as color or temperature [16]. These studies are mostly oriented to the application of VTOL UAV landing. The others are more concerned with the overall outline of the carrier, supplemented by the runway and marking lines as reference marks [17]. These studies are mostly oriented toward the application of fixed-wing UAV landing. The literature review are summarized in Table 1.

In order to estimate the relative pose of the carrier and UAV based on visual information, it is necessary to accurately segment the landing area and extract the key feature points to solve the perspective projection transformation relationship. At first, researchers mostly used threshold segmentation to extract the reference points of the landing area for its high calculation efficiency. Threshold segmentation refers to segmenting objects and backgrounds according to a certain luminance constant. Ref. [18] designed a purple-and-yellow ring-shaped auxiliary landing marker, where an adaptive threshold segmentation algorithm was used to separate the auxiliary landing marker and background according to the color phase. Ref. [19] proposed a threshold segmentation algorithm based on the HSI model, which is insensitive to illumination and color, to achieve the segmentation of the landing marker. Ref. [20] proposed an image segmentation method based on context-aware and graph-cut theory to separate landing marker from the background, aiming at solving the problem that it is difficult to eliminate background interference due to the unstable features of the marine environment. Ref. [21] presented a real-time runway detection approach based on a level set. This approach proposed a three-thresholding segmentation method to extract a runway subset from regions of interest (ROI), which was used to construct an initial level set function. Since runway relative navigation during the final approach is a special case where robust and continuous detection of the runway is required, ref. [22] presented a pure optical approach of a robust threshold marker detection for monocular cameras, which increased sensor redundancy.

Moreover, the robustness requirement of the runway detection method is increased since carrier landing scenes contain more complex shapes and texture features. Researchers have focused on robust feature description to achieve the stable detection and tracking of runways. Ref. [23] proposed a target recognition method combining SURF and FLANN to improve the recognition accuracy of landing markers in view of the complex carrier landing background and the difficulty of identifying markers under partial occlusion. In this method, a SURF descriptor was used to extract the corner features from the target candidate regions. Then, the optimal matched feature point pairs were used evaluate the correlation between the target candidate region and the image data of the ship-based marker. Ref. [24] proposed a vision-based navigation scheme for UAV autonomous carrier landing. In this paper, a spectral residual-based saliency analysis method combined with LLC-based feature learning was adopted to detect the aircraft carrier. This method fused global and local features under the spatial pyramid framework and preserved the spatial distribution relationship of features on the image to the greatest extent, which greatly improved the robustness of the detection algorithm. In this research, an effective morphological fusion method was employed for the prediction of virtual runway imagery to avoid accidents in the landing process. For this, a fusion of sensor data from DEM (digital elevation map) data, infrared images (IR) and navigation parameters was used. Ref. [25] proposed a runway detection process framework, in which a two-scale runway detector based on YOLOv3 was first used to initially detect the airport runway, and then a reclassifier based on ResNet-101 was used to improve the accuracy of the initial detection results. In [26], a deep-learning-based detection and tracking algorithm for long-distance landing was proposed to solve the problems of small and blurred landing markers in the field of view. The algorithm fused multiscale information on the basis of the SSD network model and improved the detection performance of small and medium targets by redesigning the feature extraction structure. At the same time, based on the kernel correlation filtering, the complementary filtering fusion detection algorithm was used to further reduce the detection error and improve computational efficiency. To achieve the accurate landing of UAVs in complex environments, a landing runway recognition model based on a deep learning algorithm was proposed by [27]. In this lightweight landing runway recognition model, the lightweight network Mobilenet-V3 was used to replace the underlying network structure CSPDarkNet-53 in the YOLOv4 algorithm, which meets the lightweight requirement for use on mobile devices. In order to improve the robustness of the landing marker recognition in a complex landing environment, ref. [28] proposed an improved SSD network model landing sign recognition method. This method introduced a deep residual network and feature pyramid network structure into an SSD network and adopted a ResNet101 instead of a VGG-16 network. The feature pyramid network structure was used to improve the traditional upsampling structure, and the high-level semantic information of the detection network was integrated into the low-level feature information, which improved the recognition rate and confidence of small targets and reduced the probability of misjudgment.

Furthermore, due to specific appearance characteristics of a runway, the introduction of edge features such as line distribution can improve the detection efficiency. Ref. [29] proposed a coarse-to-fine hierarchical architecture for runway detection. In this approach, runway ROIs were extracted from a whole image by the model at the coarse layer. Then, a line segment detector was applied to extract straight line segments from ROIs. Finally, edge features, such as the vanishing point and runway direction, were used to recognize the runway. Considering the center line of the runway is approximately vertical and can be determined from the Hough transform, ref. [30] adopted a runway edge detection method with horizon detection. Ref. [31] proposed a runway detection algorithm based on colors and characteristics identification. This algorithm detected the runway boundaries by selecting the appropriate Hough lines using runway characteristics and runway color. Once the runway was detected, it tracked the runway using feature matching techniques. In the tracking phase, the algorithm tracked the runway and it found out the accurate runway boundaries and threshold stripes.

Under the premise of the known width and length of the runway, ref. [32] proposed a runway detection method based on line features and a gradient scheme, aiming to solve the problems of the traditional Hough transformation, which requires much calculation time and is prone to the false detection of endpoints.

In recent years, deep-learning-based semantic segmentation algorithms have made significant progress in processing the “full” dynamics of the carrier landing environment. They can not only separate the ship background in the image, but also classify all the pixels in the image and extract high-level semantic information to realize the perception of the landing environment. Ref. [33] presented a runway detection and tracking algorithm applied to UAVs. This algorithm relied on a combination of segmentation-based region competition and a minimization of a particular energy function to detect and identify the runway edges from vision data. Ref. [34] proposed a semantic segmentation method of landing scene based on an improved ERFNet network, aiming to improve the autonomous perception ability of UAVs to the landing environment. This method introduced an asymmetric residual module and a weak bottleneck module to improve the ERFNet network model: the asymmetric residual module was designed to improve the running speed when the amount of calculation of the feature map was large; and the weak bottleneck module was used to reduce the number of parameters and reduce the loss of accuracy, which made the network more efficient for the semantic segmentation task of the carrier landing scene. Ref. [35] designed a deep convolutional neural network (CNN)-based runway segmentation method to advance autonomous capabilities and safety and achieve human-level perception for UAVs. Ref. [36] proposed a vision system design applied to UAV landing in outdoor extreme low-illumination environments without the need to apply an active light source to the marker. In this method, a model-based enhancement scheme was used to improve the quality and brightness of the onboard captured images, and a hierarchical-based method consisting of a decision tree with an associated light-weight CNN was proposed for the coarse-to-fine landing site localization, where the key information of the marker was extracted and reserved for postprocessing, such as the pose estimation and landing control.

## 3. Framework and Design

A carrier-based UAV faces extremely complex environments while landing. At the end of the landing stage, the UAV takes a fixed glide angle of 3–6° and points to the direction of the central axis of the landing area as the landing course. The longitudinal length of ships that can be used for carrier-based UAV landings is generally about 100–300 m. In this case, the airborne images mostly show the characteristics of broad range, large viewing distance and depth of field. Therefore, by comprehensively analyzing the characteristics of airborne images during the autonomous landing of a carrier-based UAV, machine vision faces the following constraints during the landing process:Limited by the spatial and temporal resolution of visual information in superlarge-scaled scenes, the feature information of some regions will be compressed or even lost.The autonomous landing of carrier-based UAVs can be regarded as an object detection and tracking problem for a moving target. In other words, the key parameters of the moving platform are known, but its global information is unknown, which is the difference with land-based landings.In addition, since the motion of the aircraft carrier is much lower relative to the airspeed of the UAV, the motion estimation of the moving target can be ignored in the scenario with a large depth of field and large viewing distance.

The basic detection framework was designed based on YOLOv5 and is shown in Figure 1; it mainly includes the following modules: a data augmentation layer, a feature extraction layer, a feature aggregation layer and a target prediction layer. In the Y0L0v5 network architecture, the smallest component **CBL** consists of a convolutional layer **conv**, a normalized layer and an activation function. This module can add the depths of feature maps with a different number of channels through tensor splicing, which expands the dimension of the tensor while keeping the width and height unchanged. In the feature extraction layer, in order to prevent the generation of excessive repeated gradient information, the feature maps of the base layer are divided and merged by introducing the **CSP** (cross stage partial network), which maintains sufficient accuracy while reducing the amount of calculation. The **Focus** module is introduced to slice the feature map to reduce the computational complexity of the algorithm. In addition, by introducing **SPP** in the feature aggregation layer, the receptive field can be increased, and the features of the context can be separated more effectively.

Since the runway is generally a regular quadrilateral with a fixed aspect ratio, the relative attitude of the UAV and the runway directly affects the geometric characteristics of the runway target in the image. Furthermore, because the height and speed of the UAV change rapidly during the landing phase, the scale of the runway in the image changes greatly and the background information changes greatly. It is worth noting that in YOLOv5, an adaptive anchor box mechanism is adopted, and it sets a priori boxes of 3 sizes for each downsampling scale, which can enrich the dataset to a certain extent. However, on the one hand, this method cannot be used for objects with irregular shapes; on the other hand, due to the limitation of rectangular boundaries, it cannot effectively deal with the rotation transformation of images. To solve this problem, this paper introduces the geometric features of the runway and the quadrilateral corners prediction information of the runway into the design of the data augmentation layer and adopts a rotation transformation to supplement the dataset. As shown in Figure 2, the yellow rectangle box is the anchor box generated by YOLOv5, while the red quadrilateral box is the anchor box generated by our method. It not only greatly enriches the background information of the dataset, but also makes the model more adaptable in highly dynamic scenes and improves the robustness of the model without increasing the inference cost.

Accurately and effectively modeling the appearance of the runway area is the basis for the fast detection of the runway in a dynamic environment. Under the YOLOv5 network framework, the appearance modeling of the target mainly relies on feature mapping. In order to enhance the representation ability of feature extraction, the feature extraction layer introduces functional submodules on the basis of the backbone network to increase the ability of the deep network to deal with special problems. The feature extraction layer introduces the **Focus** and **CSP** modules on the basis of the Darknet53 network to replace the three convolutional layers, which reduces the amount of network parameters and enhances the representation ability of the feature map. Since different network layers have different emphasis on target information extraction, the feature pyramid is introduced in the feature aggregation layer to effectively utilize the global and local information of the image, thereby further improving the robustness of the network model. Considering the large variation of the runway scale in the image during the landing process, the feature aggregation layer adopts **SPP** to enhance the ability of the network feature fusion, which enriches the expressive ability of the feature mapping.

## 4. Corner Prediction

On the premise that the airborne camera captures the runway and tracks it robustly, as the relative distance between the UAV and the runway gradually decreases, the texture information of the runway becomes clear. At that time, the reference corners of the runway can be effectively extracted and used as reference objects to construct *n* pairs of coordinates of two-dimensional points in the image coordinate system and coordinates of the corresponding three-dimensional points in the world coordinate system. The subsequent state estimates are made on this basis. In the image captured by the airborne forward-looking camera, as illustrated in Figure 3, the area of the runway generally has the following characteristics: the area of the runway is different from the background, there are marking lines in the middle at the edge of the runway and the area of the runway is presented as a convex quadrilateral. The specific geometry is related to the relative attitude between the UAV and the runway.

The traditional corner prediction method includes two stages: target segmentation and corner fitting. First, the runway is segmented from the image, and then the corners of the runway are extracted from the segmented runway target. This method is highly dependent on the segmentation results of the first stage. However, the traditional target segmentation algorithm using manual features has the disadvantages of a slow calculation speed, low accuracy and poor generalization ability. The proposed runway corner prediction method converts the traditional two-stage method into a coordinate regression problem of corner features, which can not only ensure a high-precision prediction of corner points, but also greatly improve the calculation speed. It can be seen that YOLOv5 has the advantages of having less parameters, less training time and a fast inference, and can effectively deal with the problem of runway detection in the high-speed dynamic scene of a UAV landing.

### 4.1. Calculation for Intersection over Union

The intersection over union (IoU) of two convex quadrilaterals was calculated based on the geometric method. The basic idea of the geometry-based convex quadrilateral IoU calculation method is as follows: first, the vertex coordinates of the intersection graph of two convex quadrilaterals are obtained from their geometry, and then the area of the intersection graph is obtained by a vector cross product. The time complexity of the algorithm is low because it only needs to operate according to the vertex coordinates of the two convex quadrilaterals.

A vertex is selected as the starting point, and the convex polygon is divided into triangles, as presented in Figure 4. Then, the formula for calculating the area of a convex polygon can be expressed as:(1)$$S_{n}=\sum_{i=1}^{n-1}S_{i}$$
where *n* is the number of vertices of the convex polygon and *S* is the area of each triangle.

Without loss of generality, the intersection area of the two convex quadrilaterals $T_{1}$ and $T_{2}$ forms a convex polygon *L*, as shown in Figure 5. The vertices of polygon *L* are generally divided into two types, among which vertices A, B and C are the vertices of the original convex quadrilateral, which are, respectively, included in the opposite area, while vertices D and E are the intersections of the sides of the two convex quadrilaterals.

**Property** **1.**
*For three points A, B and C which are coplanar and noncollinear, the order of them can be judged by the right-hand theorem of the vector cross product by constructing the vectors $\vec{AB}$ and $\vec{AC}$. If the cross product $\vec{AB}\times \vec{AC}>0$, the three points A, B and C are arranged in a clockwise direction, else if the vector cross product $\vec{AB}\times \vec{AC}<0$, the three points are in a counterclockwise direction.*


According to Property 1, the judgment mechanism of whether any point *P* is located inside a convex quadrilateral can be expressed as follows: construct two vectors from the point *P* to the two vertices of the convex quadrilateral in turn and perform the cross product operation in sequence. If the signs of the results are the same, the order of vertices must be clockwise or counterclockwise. For example, if $\vec{PA}\times \vec{PB}>0$, $\vec{PB}\times \vec{PC}>0$, $\vec{PC}\times \vec{PD}>0$, $\vec{PD}\times \vec{PA}>0$, the point *P* is inside the convex quadrilateral, as shown in Figure 6a. Otherwise, point *P* is outside the convex quadrilateral, as shown in Figure 6b.

For the solution of the position of the intersection of the two sides, as shown in Figure 7, first take a point *O* so that $|\vec{AQ}\left|=\right|\vec{PO}|$ and $\vec{AQ}\times \vec{PO}=0$ are satisfied. Then, the following relationship can be expressed as:(2)$$\frac{\vec{AB}}{\vec{AQ}}=\frac{S_{\Delta APC}}{S_{\Delta POC}}=\frac{|\vec{PA}\times \vec{AC}|}{|\vec{PO}\times \vec{CP}|}=\frac{|\vec{PA}\times \vec{AC}|}{|\vec{AQ}\times \vec{CP}|}$$

As shown in the figure, the concept of the smallest closed box in the GIoU (generalized intersection over union) is introduced in the calculation of the IOU. On this basis, the following improved method is proposed: when taking the smallest closed box that contains these two intersecting convex quadrilaterals, no additional vertices are added to form a convex polygonal smallest closed box, that is, it represents the minimum generalized intersection over union, as shown in Figure 8. The above IoU solution method is defined in Algorithm 1.
**Algorithm 1** Solution for sGIoU of two convex quadrilaterals**Input:** Coordinate information of two convex quadrilaterals with intersecting part of *L*
**Output:** Loss of the minimum generalized intersection
1:Solve the coordinates of intersection convex polygon vertices: *A*, *B*, *C*, *D* and *E*2:Solve the area of two convex polygons and intersection: $S_{1}$, $S_{2}$ and $S_{L}$3:The intersection over union is:
(3)$$IoU=\frac{S_{L}}{\left|S_{1}+S_{2}-S_{L}\right|}$$4:The loss of minimum generalized intersection over union is:
(4)$$L_{sGIoU}=1-IoU+\frac{\left|S_{G}-S_{1}\cup S_{2}\right|}{S_{G}}$$
where $S_{G}$ is the smallest convex polygon enclosing box containing these two convex quadrilaterals.


### 4.2. Loss Function with Geometric Constraints

The corners of the prediction target box care less about the center point and length and width of the object box, but focus on the four corners of the runway. As shown in Figure 9, the offset loss between the corners of the predicted box of the convex quadrilateral target and the ground-truth box can be expressed as:(5)$$L_{offs.}=l^{obj}\sum_{i=0}^{n}\left[{\left(x_{i}-{\hat{x}}_{i}\right)}^{2}+{\left(y_{i}-{\hat{y}}_{i}\right)}^{2}\right]$$
where $\left(x_{i},y_{i}\right)$ represents the coordinate of the *i*th corner of the ground-truth box, $\left({\hat{x}}_{i},{\hat{y}}_{i}\right)$ represent the coordinates of the *i*th corner of the predicted box, $l^{obj}$ represents the effectiveness of the convex quadrilateral box for object prediction and *n* is the number of corners.

In order to characterize the spatial sequence information of the corners, the four corners of the prediction frame are sequenced according to the following rules: the first vertex in the upper left is set as the starting point, and all the corners are sorted clockwise. Therefore, the geometric constraints between the corners can be expressed as:(6)$$\left\{\begin{array}{c} x_{1}⩽x_{2}\\ x_{4}⩽x_{3}\\ y_{3},y_{4}\geq y_{1},y_{2} \end{array}\right.$$

Then, the loss of the corner sequence constraint can be expressed as: (7)$$L_{seqc.}=\left[max\left(y_{1}-y_{3},0\right)+max\left(y_{2}-y_{3},0\right)+max\left(y_{1}-y_{4},0\right)+max\left(y_{2}-y_{4},0\right)+max\left(x_{1}-x_{2},0\right)+max\left(x_{4}-x_{3},0\right)\right]/6$$

In summary, the loss function of the corner prediction can be expressed as:(8)$$Loss=L_{sGIoU}+L_{offs.}+L_{seqc.}$$

## 5. Experimental Setup and Evaluation

### 5.1. Setup and Training

The benchmarking datasets were generated by FlightGear (ver. 2020.3), which is one of the most realistic flight simulators developed based on the OpenSceneGraph and features detailed worldwide scenery modeling and a flexible and open aircraft modeling system. The landing scene of the carrier-based UAV was constructed in a marine scene with the help of Nimitz (CVN-68). As shown in Figure 10, the carrier-based UAV flew from 1.6 nm (about 3 km) away and approached the aircraft carrier at 1700 ft (about 0.5 km) above sea level. The aircraft realigned to the carrier during the whole flight and then guaranteed the camera was focusing on the deck at the image center. The ground truth of the trajectory was recorded by FlightGear.

First, the geometric model of camera imaging needed to be established, that is, the internal parameters of the virtual camera that generated the simulated image needed to be calibrated. We constructed an 8×8 checkerboard with a size of 20 m × 20 m via AC3D (ver. 9.0.22) and ported this 3D model to the landing scene. The onboard camera captured checkerboard images from different angles to calibrate the camera parameters, as shown in Figure 11. The camera model parameters such as the camera internal parameter matrix, distortion sparse and field of view obtained by the checkerboard calibration method are summarized in Table 2.

The vision-based estimation of the relative pose between the UAV and the carrier consists in solving the perspective transformation. In another word, it is necessary to solve the coordinate transformation relationship of the three-dimensional geometric position of points in the global coordinate and their corresponding points in the image coordinate. We defined the center point of the FLOLS of the Nimitz as the origin of the coordinate system. The global coordinate was established with the direction pointing to the stern along the port side as the *x*-axis, the direction perpendicular to the *x*-axis pointing to the starboard side along the ship surface as the *y*-axis and the direction perpendicular to the xy plane pointing to the sky as the *z*-axis. The definition of the global coordinates and three views of the Nimitz aircraft carrier model are shown in Figure 12, which are the side view, front view and vertical view in sequence. In the vertical view, the runway area was defined as an area surrounded by four points *A*, *B*, *C* and *D*. According to the 3D model parameters of the aircraft carrier, the coordinates of points *A*, *B*, *C* and *D* in the coordinate system could be calculated as: (−73.90, 0, 0), (148.50, 33.60, 0), (145.50, 58.70, 0), (−101.20, 22.20, 0).

All the experiments were carried out on an AMD CPU Ryzen 7 5800H @ 3.2 GHz processor 16 GB RAM and NVIDIA GeForce RTX 3060 Laptop GPU. We implemented the networks using the publicly available framework Pytorch 1.21.1 and trained it using CUDA 11.6. All the generated images were divided into three datasets: the training set contained 2400 images, while the validation set and testing set had 300 images, respectively, (the ratio of 8:1:1). Among them, the training set and validation set were used for model training, and the testing set was used to test the performance of the method. During this training, the size of the images was adjusted to 1280×800. The hyperparameters of the training model, which were mainly used to set the learning rate, loss function and data enhancement parameters, are presented in Table 3. The training results, R curve and PR curve are shown, respectively, in Figure 13 and Figure 14a,b.

### 5.2. Performance Evaluation

The mean average precision (mAP) and mIoU (mean IoU) were utilized as qualitative evaluation metrics for the accuracy of the runway detection, which characterized the accuracy of the algorithm. The mAP can be expressed as:(9)$$mAP=\int_{0}^{1}\frac{N_{TP}}{N_{TP}+N_{FP}}dr$$
where $N_{TP}$ represents the number of true positive prediction results, $N_{FP}$ represents the number of false positive prediction results and *r* represents the recall of prediction results.

Furthermore, the mIoU can be expressed as:(10)$$mIoU=\frac{1}{n}\sum \frac{S_{pred}\cap S_{gt}}{S_{pred}\cup S_{gt}}$$
where $S_{pred}$ represents the area of the prediction, $S_{gt}$ represents the area of the ground truth and *n* is the number of samples.

To verify the effects of the two improved methods proposed in this paper, we conducted an ablation study. The results of the ablation study are shown in Table 4. According to the results of the ablation experiment, the improved data augmentation method improved mAP and mIoU by 0.1% and 4.5%, respectively, without affecting Fps. Moreover, the corner prediction method greatly improved the detection effect, in which Fps decreased slightly, and the mAP and mIoU increased by 0.6% and 18.3%, respectively. In Figure 15, the blue rectangular box represents the prediction result of the baseline method, and the red quadrilateral box represents the prediction result of the proposed method.

We also conducted experiments on landing scenes in different weather conditions, including sunny weather (with visibility of 20 km) and foggy weather (with visibility of 2 km), as shown in Figure 16 and Figure 17. It should be pointed out that only the prediction results within its effective scope are presented since the effectiveness of machine vision was limited by the visibility. The results shown in Table 5 indicate that the proposed method can achieve a high detection precision in the visual range of machine vision under different weather conditions.

To verify the effectiveness of the proposed method and other algorithms, we conducted the following comparative trials. From three aspects, detection method, detection framework and detection strategy, three methods were selected to compare with the method proposed in this paper. In the following comparative experiments, the images under sunny conditions were used for testing. Table 6 shows the comparison results and performance. The method proposed by [24] is a typical region detection method which has a good accuracy performance but a poor computational efficiency. In this method, only the rough runway area could be detected, but the precise runway position could not be given. The results also showed that the precision of the proposed method improved by nearly 1% compared with the baseline method of YOLOv5, the degree of coincidence with the appearance of the runway also improved by about 18%, and the computational efficiency was almost not reduced. This pair of experiments showed the superiority of the method proposed in this paper, achieved by improving the data augmentation method and introducing the quadrilateral corners’ prediction mechanism. In order to evaluate the performance of the proposed framework, the third comparison method adopted the YOLOv5 baseline method and edge extraction method proposed by [32]. This method could realize a precise detection of the landing site, but the detection performance was poor because of the lack of a clear geometric description of the target.

### 5.3. Precision Evaluation

Furthermore, a reprojection error was adopted as a quantitative evaluation metric to assess the precision of state estimation. Suppose a principal point in the three-dimensional space of the global coordinate system is $P_{i}={\left[X_{i},Y_{i},Z_{i}\right]}^{T}$ and its coordinate in the normalized projection plane is $p_{i}={\left[u_{i},v_{i},1\right]}^{T}$. Then, the transformation relationship between the global coordinate and the projection plane can be expressed as:(11)$$s_{i}p_{i}=K\exp\left(\hat{\xi }\right)P_{i}$$
where $s_{i}$ represents the relative depth, *K* represents the parameters of the camera model and ξ is the mapping transformation represented by a Lie algebra.

As shown in Figure 18, the noise of the observation point and the prediction error cause a reprojection error between the projection of the observation point and the principal point on the image plane. Thus, the reprojection error can be expressed as:(12)$$e=p_{i}-\frac{1}{s_{i}}K\exp\left(\hat{\xi }\right)P_{i}$$

Figure 19 shows the relationship between the reprojection error of the corner point prediction and the flight distance during the landing phase. It can be inferred that the reprojection error of the corner point prediction was less than 0.002, within a range of about 2.5 km from the aircraft carrier. That is, for commonly used high-resolution images, the accuracy of state estimation could be within two pixels, which is very impressive.

## 6. Conclusions

Aiming at improving the autonomous landing capability of carrier-based UAVs, this paper proposed a monocular-vision-based state estimation method, which was dedicated to meet the requirements of accuracy and robustness. The framework was designed with four layers: a data augmentation layer, a feature extraction layer, a feature aggregation layer and a target prediction layer. In this framework, the improvement of the data augmentation achieved more robust detection performance. The introduction of a corner prediction mechanism based on geometric features enabled a more precise landing field prediction, which fit the runway appearance better. In simulation experiments, we developed datasets suitable for precision UAV landing simulations based on monocular vision. Simulation results showed that the reprojection error of the corner point prediction was less than 0.002 within a range of about 2.5 km, and the accuracy of state estimation could be within two pixels, which yielded a higher precision and better performance of state estimation. By improving the framework, our method solved the problem that the YOLOv5 detection model could not be adapted to the task of state estimation during carrier-based UAVs landing since it could not obtain critical features such as the edges or corners of the runway directly.

In the future work, we will devote ourselves to the research on landing carrier-based UAVs by developing autonomous control technology, based on the work from this paper. In the study of state estimation during carrier-based UAV landing, we will fix the shortcoming that scale information is missing under monocular vision, through multimodal information fusion methods, such as introducing the navigation information of INS and GNSS. In the next stage, we will test the algorithm by using real scene images instead of virtual scenes. Furthermore, a hardware-in-the-loop simulation will also be conducted to assist theoretical research and reliability verification in the future. At that time, we will propose a robust and intelligent fusion strategy and provide theoretical innovations for the development of autonomy and intelligence in UAVs. 

## Figures and Tables

**Figure 1 sensors-22-08385-f001:**
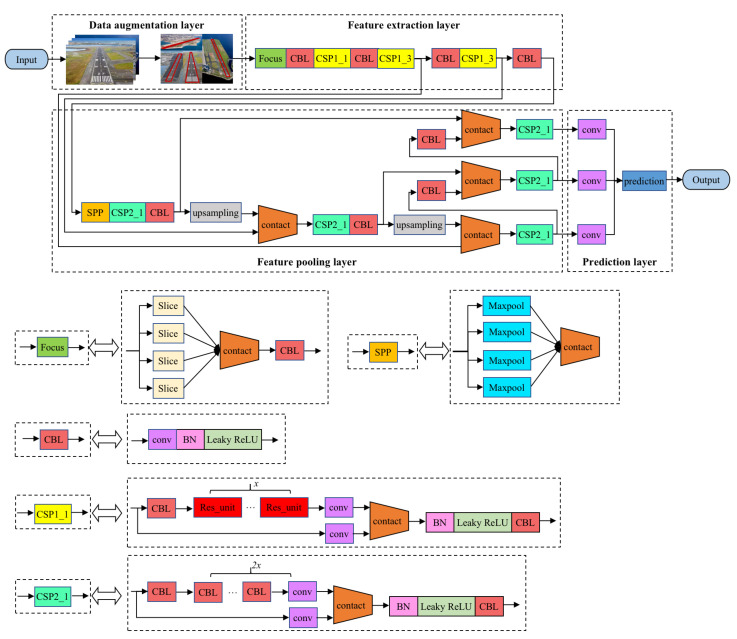
Framework of runway detection.

**Figure 2 sensors-22-08385-f002:**
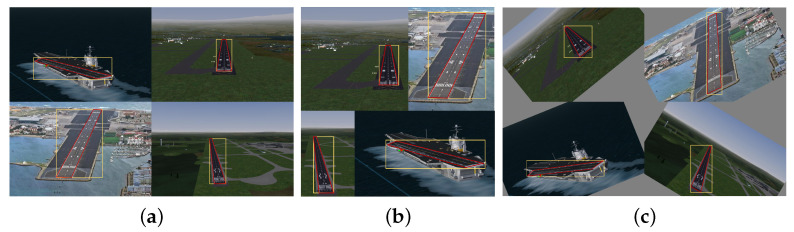
Improvement of data augmentation. (**a**) Images of datasets, (**b**) data augmentation by YOLOv5, (**c**) ours.

**Figure 3 sensors-22-08385-f003:**
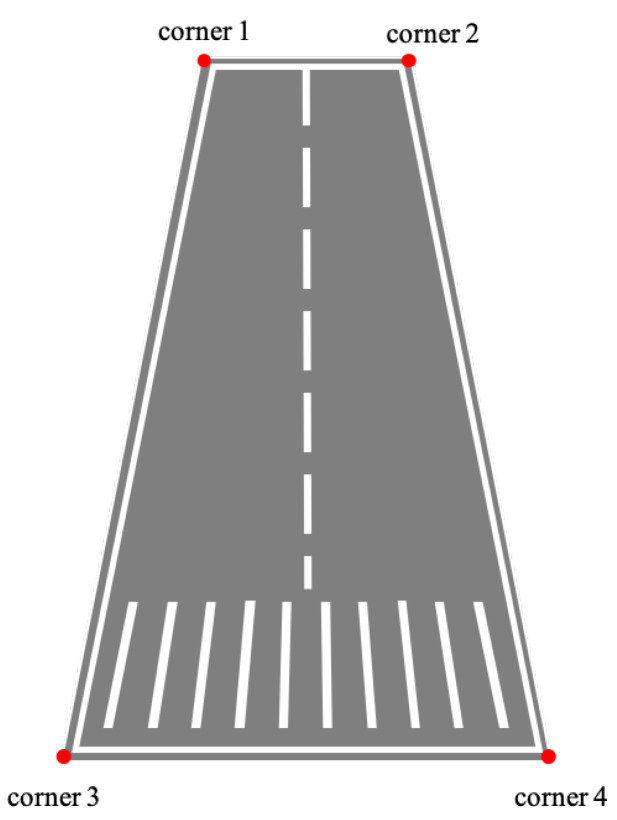
Typical geometry of runway captured by the airborne forward-looking camera.

**Figure 4 sensors-22-08385-f004:**
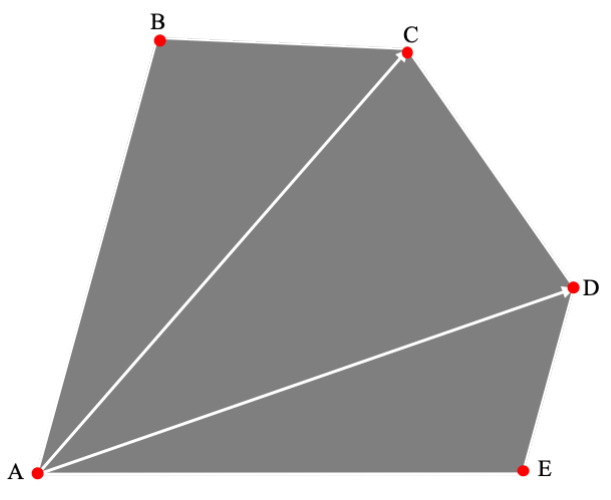
Schematic diagram of convex polygon area calculation.

**Figure 5 sensors-22-08385-f005:**
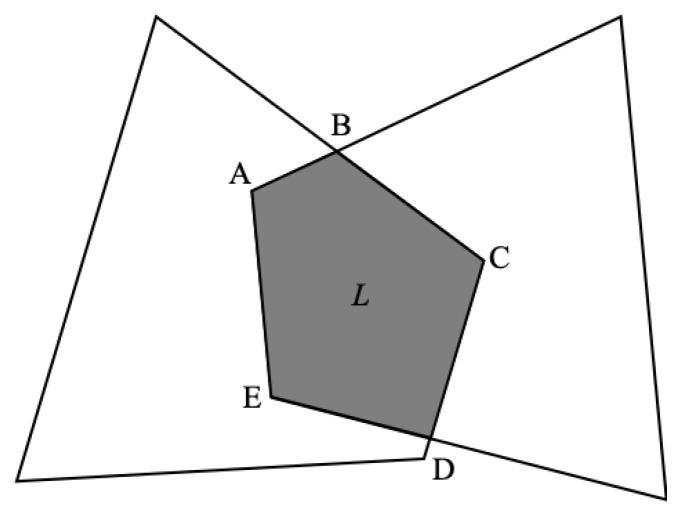
Schematic diagram of the intersection area of the two convex quadrilaterals.

**Figure 6 sensors-22-08385-f006:**
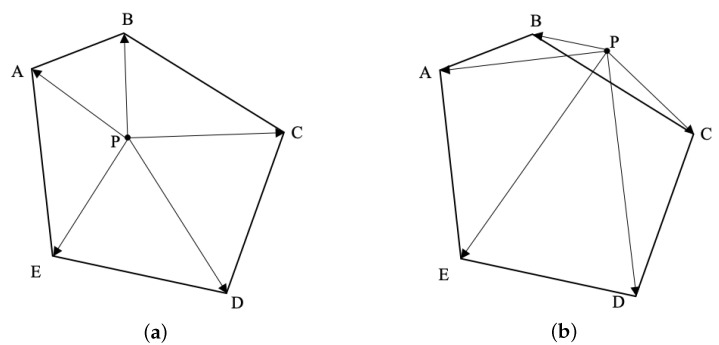
Schematic diagram of the judgment mechanism of whether any point *P* is located inside a convex quadrilateral.

**Figure 7 sensors-22-08385-f007:**
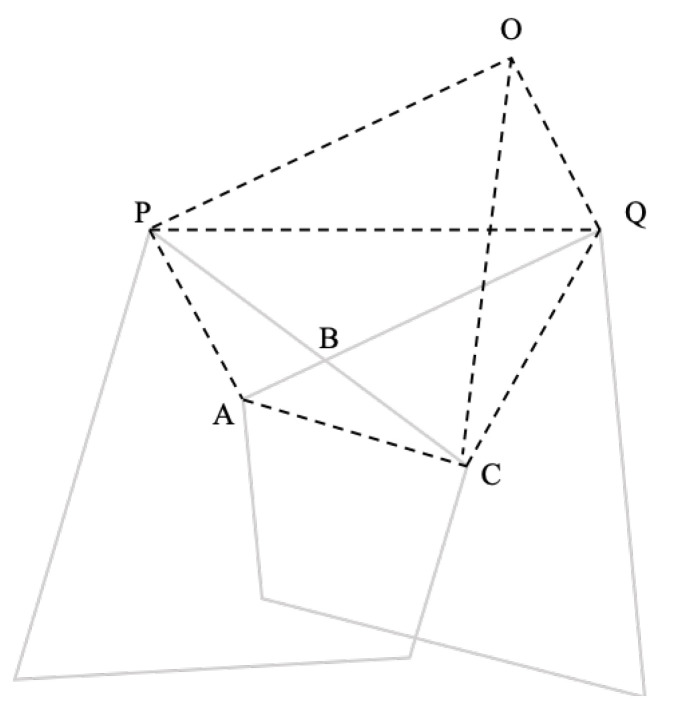
Schematic diagram of solution for intersection position of two sides.

**Figure 8 sensors-22-08385-f008:**
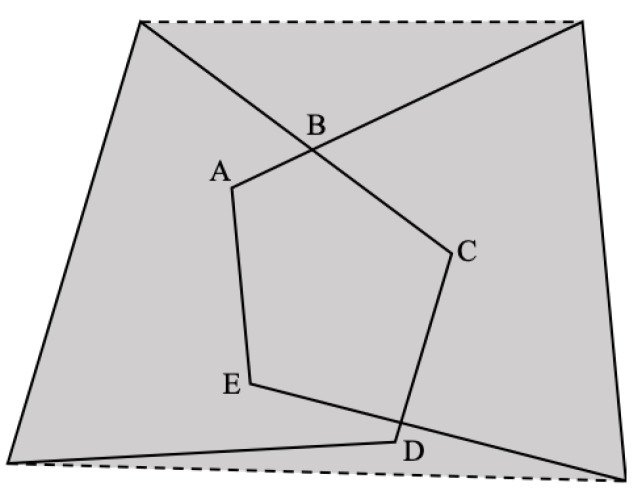
Schematic diagram of the smallest closed box containing the two convex quadrilaterals.

**Figure 9 sensors-22-08385-f009:**
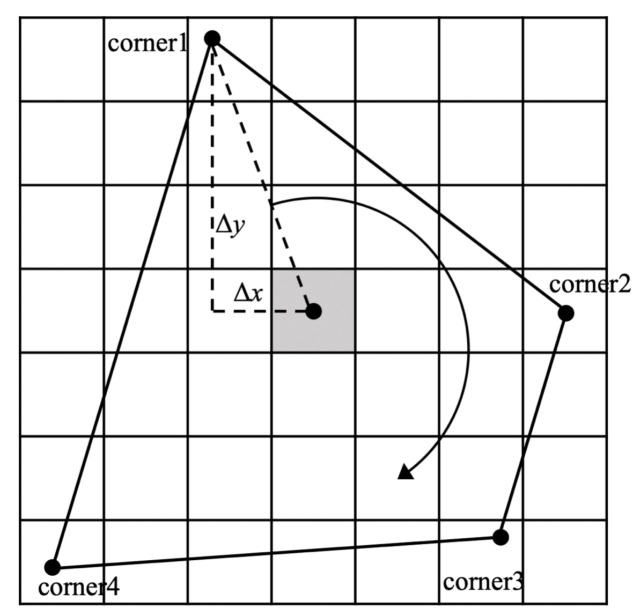
Schematic diagram of corner prediction.

**Figure 10 sensors-22-08385-f010:**
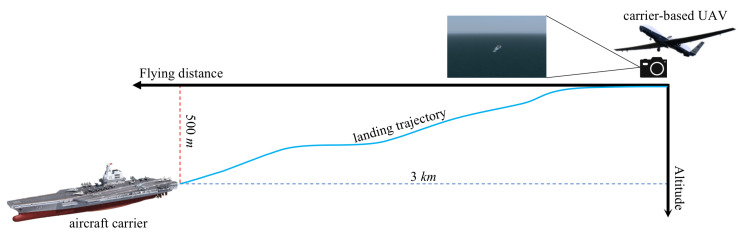
Schematic diagram of carrier-based UAV landing stage.

**Figure 11 sensors-22-08385-f011:**
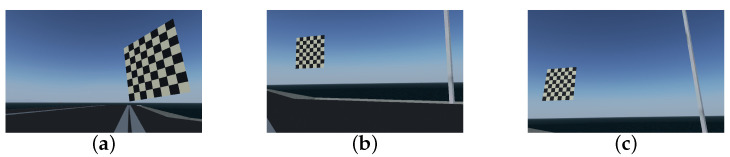
Checkerboard calibration. (**a**) Sample 1 of calibration image. (**b**) Sample 2 of calibration image. (**c**) Sample 3 of calibration image.

**Figure 12 sensors-22-08385-f012:**
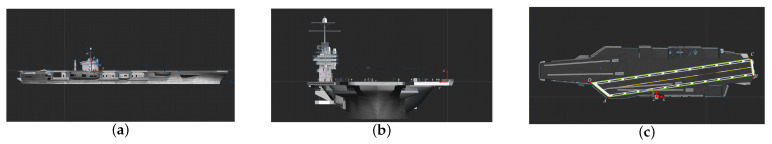
Definition of global coordinate and three views of the Nimitz. (**a**) The side view of Nimitz. (**b**) The front view of Nimitz. (**c**) The top view of Nimitz.

**Figure 13 sensors-22-08385-f013:**
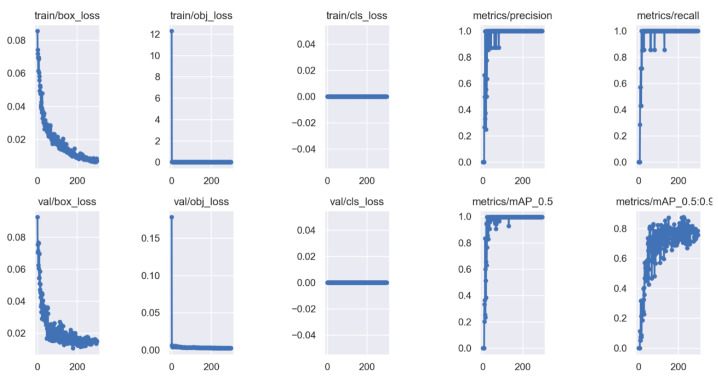
Results of data training.

**Figure 14 sensors-22-08385-f014:**
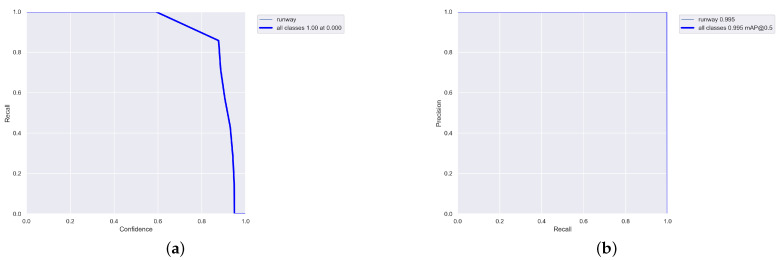
R curve and PR curve of training results. (**a**) R curve. (**b**) PR curve.

**Figure 15 sensors-22-08385-f015:**
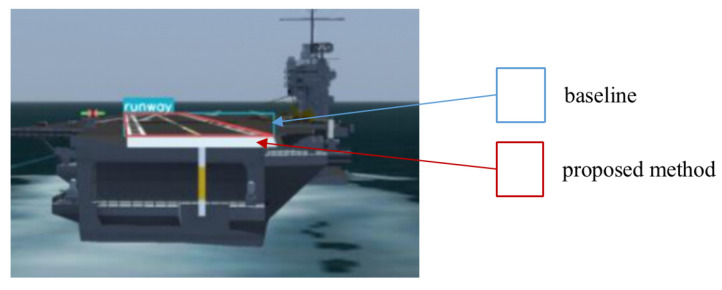
Schematic diagram of the comparison of runway prediction results of different methods.

**Figure 16 sensors-22-08385-f016:**
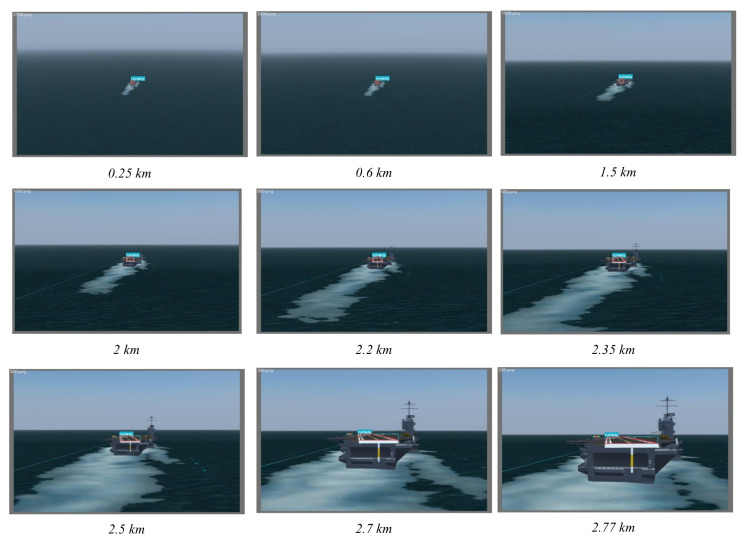
Results of runway prediction corresponding to flight distance under sunny weather.

**Figure 17 sensors-22-08385-f017:**
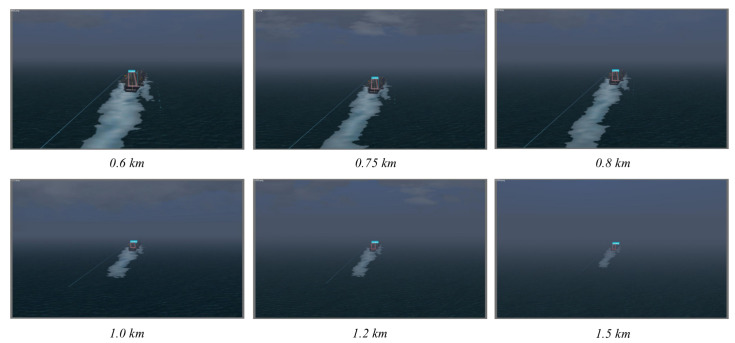
Results of runway prediction corresponding to flight distance under foggy weather.

**Figure 18 sensors-22-08385-f018:**
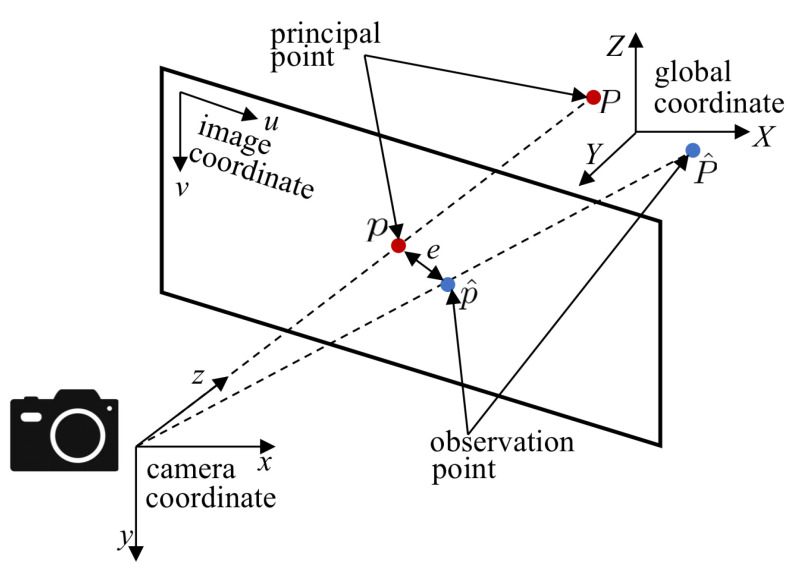
Schematic diagram of reprojection error corresponding to flight distance.

**Figure 19 sensors-22-08385-f019:**
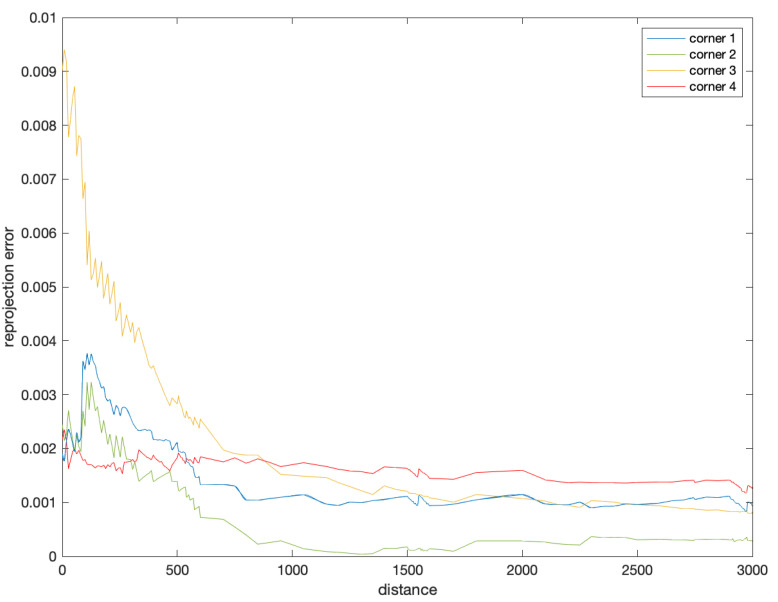
Reprojection error of corner prediction.

**Table 1 sensors-22-08385-t001:** Summary of literature review.

Approach	Characteristics	Citation
Threshold segmentation	High computing efficiency but limited detection accuracy.	[18,19,20,21,22]
Region detection	It has a high detection robustness and accuracy. However, region detection methods are too complicated and time-consuming, and only a rough detection of the runway can be achieved.	[23,24,25,26,27,28]
Edge detection	This method achieves a more precise runway detection performance compared to region detection. It is a typical two-stage detection method, including region detection and edge feature extraction. Therefore, it has the disadvantage of a low computational efficiency.	[29,30,31,32]
Semantic segmentation	It has high-precision detection performance and can realize the perception of the landing environment. However, the continuity of the runway boundary detection is insufficient. Moreover, semantic segmentation methods are complicated and time-consuming.	[33,34,35,36]

**Table 2 sensors-22-08385-t002:** Parameters of camera model.

Parameters	Value
FOV	70${}^{\circ }$
Resolution	3360 × 2100
Intrinsic parameters	$\left[\begin{array}{ccc} 2.4084\times 10^{3} & 0 & 0\\ 0 & 2.4114\times 10^{3} & 0\\ 1.6614\times 10^{3} & 1.0471\times 10^{3} & 0 \end{array}\right]$
Radial distortion	$\left[0.0035,0.0089,-0.0147\right]$
Tangential distortion	$\left[6.5915\times 10^{-4},4.7761\times 10^{-4}\right]$

**Table 3 sensors-22-08385-t003:** Hyperparameters of detection model.

Hyperparameters	Value	Hyperparameters	Value
Initial learning rate	0.0001	Optimizer weight decay	0.0005
Momentum	0.937	Number of iterations	300
Learning batch size	4	Loss gain for sGIoU	0.05
Loss gain for classifier	0.5	Loss gain for geometry	1.2
Threshold for training	0.2	Threshold for sGIoU	0.15
Smooth scope for loss function	0.02		

**Table 4 sensors-22-08385-t004:** Results of ablation study.

Baseline	Data Augmentation	Corner Prediction	Prediction Box	mAP	mIoU	Fps
✓			Rectangle	0.989	0.771	66.7
✓	✓		Rectangle	0.99	0.825	66.6
✓	✓	✓	Quadrilateral	**0.995**	**0.954**	**64.5**

**Table 5 sensors-22-08385-t005:** Results under different weather conditions.

Landing Scene	Prediction Box	mAP	mIoU	Fps
Sunny weather	Quadrilateral	**0.995**	**0.954**	**64.5**
Foggy weather	Quadrilateral	**0.824**	**0.876**	**64.5**

**Table 6 sensors-22-08385-t006:** Comparison results.

Method	Prediction Box	mAP	mIoU	Fps
Method of [24]	Rectangle	0.983	0.728	0.20
YOLOv5 [37]	Rectangle	0.989	0.771	66.7
YOLOv5 + LBP [32]	quadrilateral	0.942	0.699	14.9
Ours	Quadrilateral	**0.995**	**0.954**	**64.5**

## Data Availability

Not applicable.

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
