# Peer review of "Monocular-Vision-Based Precise Runway Detection Applied to State Estimation for Carrier-Based UAV Landing"

_sensors, 2022, doi:10.3390/s22218385_

Round 1

Reviewer 1 Report

1.The introduction is long and lacks logic. Please reorganize the introduction

2.What is the research significance of this article?

3.There is little literature in the past three years, and the latest research should be cited.

4.It is recommended to supplement ablation test

5.Inadequate innovation points of this article

Author Response

Dear Editor and Reviewers,

On behalf of my co-authors, we are very grateful to you for giving us an opportunity to revise our manuscript. Thank you very much for your comments which are valuable and helpful for revising and improving our study. We have studied comments carefully and tried our best to revise our manuscript according to the comments. The following are responses and revisions I have made in response to the questions and suggestions on an item-by-item basis. In the revised manuscript, the highlighted part are the contents we added and the revised pictures and tables, and the red strikethrough part are the contents we deleted. Please see the attachment.

We hope that this revised version answers reviewers’ questions and concerns, and meets both your and reviewers’ expectations and the standards for publication in the Sensors.

Special thanks to you for your good comments and suggestions.

Best regards,

Ning Ma, Xiangrui Weng, Yunfeng Cao and LinBin Wu

Reviewer 2 Report

In this paper, an estimation method for autonomous landing of fixed-wing UAVs is proposed. The YOLOv5 convolutional network is improved to predict the corner points of the runway. Some innovative works are presented in this paper, but some major revisions are required.

(1) Two improvements are made to improve the YOLOv5 convolutional network. The experimental part lack the analysis (i.e., ablation study) of two improvements to demonstrate their performance.

(2) You should compare your method with more deep learning methods proposed recently. In addition, I do not see the reference number of the YOLOv5 network.

(3) Only the simulated experiment is provided. You should also perform the experiment on real data.

(4) The figure 13 and figure 14 are unclear. The fonts are too small.

(5) There are something wrong with the sentences, e.g., line 313, 275.

Author Response

(The authors gave the same response as above.)

Reviewer 3 Report

The paper entitled „Monocular Vision based Precise Runway Detection Applied to State Estimation for Carrier-based UAV Landing” proposed an end-to-end state estimation method for the autonomous landing of carrier-based  unmanned aerial vehicles based on monocular vision.

In the following, some issues should be clarified.

1. Please add in the abstract section the deep learning tools used and their aim.

2. The introduction section is hard to follow, please split this section into two sections Introduction and Related work. 

3. The corner regression methods are not described.

4. Please add a table with the hyperparameters of the YOLOv5 method.

5. FlightGear generated the datasets, please add how many images were used in your study and the percentage used for training, and the image attributes. 

6. In the conclusion section the authors said that "By improving the framework, our method solves the problem that YOLOv5 couldn’t be applied to state estimation for carrier-based UAVs Landing” but YOLOv5 family of object detection architectures and models pertained to the COCO dataset. It is unclear if the YOLOv5 has pertained to your dataset.

7. Please add the information if your method has good results when the images generated by FlightGear continue artifacts as smoke or fog ( the images are acquisitions in different weather conditions)

8. The quality of figures 13, 14, and 18 should be improved.

9. The study is based on 30 new references and the continuation of the study [14] is appreciated.

Author Response

(The authors gave the same response as above.)

Round 2

Reviewer 1 Report

修订后,介绍更合乎逻辑,创新点更能体现

Reviewer 2 Report

The fontsize of Figure 14 is still too small